# Myc, Oncogenic Protein Translation, and the Role of Polyamines

**DOI:** 10.3390/medsci6020041

**Published:** 2018-05-25

**Authors:** Andrea T. Flynn, Michael D. Hogarty

**Affiliations:** 1Children’s Hospital of Philadelphia, Philadelphia, PA 19104, USA; 2Perelman School of Medicine, University of Pennsylvania, Philadelphia, PA 19104, USA; hogartym@email.chop.edu

**Keywords:** polyamines, *MYC*, protein synthesis in cancer, neuroblastoma

## Abstract

Deregulated protein synthesis is a common feature of cancer cells, with many oncogenic signaling pathways directly augmenting protein translation to support the biomass needs of proliferating tissues. MYC’s ability to drive oncogenesis is a consequence of its essential role as a governor linking cell cycle entry with the requisite increase in protein synthetic capacity, among other biomass needs. To date, direct pharmacologic inhibition of MYC has proven difficult, but targeting oncogenic signaling modules downstream of MYC, such as the protein synthetic machinery, may provide a viable therapeutic strategy. Polyamines are essential cations found in nearly all living organisms that have both direct and indirect roles in the control of protein synthesis. Polyamine metabolism is coordinately regulated by MYC to increase polyamines in proliferative tissues, and this is further augmented in the many cancer cells harboring hyperactivated MYC. In this review, we discuss MYC-driven regulation of polyamines and protein synthetic capacity as a key function of its oncogenic output, and how this dependency may be perturbed through direct pharmacologic targeting of components of the protein synthetic machinery, such as the polyamines themselves, the eukaryotic translation initiation factor 4F (eIF4F) complex, and the eukaryotic translation initiation factor 5A (eIF5A).

## 1. Introduction

The polyamines (putrescine, spermidine, and spermine) are small organic cations required in nearly all organisms, from bacteria to mammals, to support cell growth and proliferation [1]. Polyamine abundance is increased in many human cancers, as the polyamine synthetic pathway is a direct downstream target of several oncogenes, including the *MYC* family [2,3,4,5]. The *MYC* proto-oncogenes (*MYC*, *MYCN*, *MYCL*) have been extensively studied since their discovery in the early 1980’s [6] and they continue to be of great interest as the most commonly deregulated genes in human cancer [7]. *MYC* genes encode highly homologous helix-loop-helix leucine zipper transcription factors, and MYC overexpression correlates with aggressive tumor behavior and poor prognosis in a wide array of cancers [8,9,10,11,12]. MYC plays a central role in creating the biomass necessary to drive cell progression, including significant increases in protein translation. Deregulated protein synthesis is a common feature of human cancers, and recent work has led to a more complete understanding of how qualitative and quantitative alterations in translation control are sentinel to cancer development, maintenance, and progression [13,14].

While MYC oncoproteins provide an attractive target for cancer therapy because of their frequent somatic activation and the addiction that MYC-driven tumors have to these oncoproteins (reviewed in Gabay et al. [15]), they have been difficult to directly inhibit pharmacologically [7,16]. Attempts to target MYC using microRNAs or antisense RNAs, or small molecules that interfere with MYC–MAX dimerization, DNA binding or stability are under investigation. Alternatively, therapeutically targeting the principal oncogenic outputs of hyperactivated MYC, such as those driving protein synthesis, may provide an alternative anti-cancer approach. The polyamine synthetic pathway is one such pathway [17]. In this review, we focus on the intersection of MYC-driven translation, the effects of polyamine depletion on protein translation, and the cellular dependencies that exist at this juncture as potential avenues of therapeutic intervention.

## 2. MYC-Driven Oncogenesis

*MYC* family of proto-oncogenes (*MYC*, *MYCN*, *MYCL*) encode highly homologous transcription factors whose activities are tightly regulated in normal cells but frequently deregulated through translocations, amplifications, or alterations of upstream signaling pathways in approximately half of all human cancers [7,18]. MYC genes function at a central node of cellular signaling to link the commitment to enter the cell cycle (stimulated by diverse inputs to MYC) with the requisite biomass production and energetics necessary to do so [19,20,21]. Indeed, MYC overexpression alone is sufficient to push quiescent cells through the cell cycle [22], while depleting cells of MYC greatly impedes protein translation and markedly reduces proliferation [23]. This non-redundant role in the control of cell proliferation also imbues MYC with significant oncogenic potential when it is deregulated.

Capitalizing on this, many mouse models of cancer have been developed using tissue-specific transgenic overexpression of *MYC* genes, providing insight into the role of MYC in cancer. Such models include the *Eμ-MYC* model of B cell leukemia/lymphoma, the *TH*-*MYCN* model of neuroblastoma, the Lo-MYC and Hi-MYC models of prostate cancer, and the involucrin-c-mycER skin cancer model [24,25,26,27]. These and other models provide important tools to dissect the molecular mechanisms by which MYC overexpression drives oncogenesis within a tissue and a platform for identifying cooperating lesions and testing therapeutic agents for the prevention or treatment of these malignancies.

### MYC and Protein Synthesis

MYC is a promiscuous transcription factor regulating thousands of target genes through both canonical target gene promoter binding and accumulation at promoters of actively transcribed genes, leading to transcriptional amplification [28,29,30,31]. Yet, much of MYC’s output supports biomass production. Since 55–75% of the dry biomass of a cell is protein or involved in protein processing [32], much of MYC’s output involves the regulation of genes that support translation: ribosomal proteins, rRNAs, tRNAs, and initiation and elongation factors [21]. Transcriptome analyses in cells in which MYC expression was modulated from null through supraphysiological [33] and in diverse cell types [34] confirm MYC’s primordial function in regulating ribosome biogenesis and protein synthesis. This enrichment in the regulation of ribosomal proteins and rRNA, along with genes involved in protein synthesis and turnover, has been similarly shown for *MYC*-driven cancers such as neuroblastoma [35,36]. Genome-wide, MYC binding at the promoters of genes involved in ribosome biogenesis or protein synthesis accounts for roughly the same number of binding events observed for genes involved in cell cycle progression, highlighting the importance of proteogenesis in MYC-driven oncogenesis [37].

Protein processing not only features prominently downstream of MYC signaling, but also leads to a MYC dependency in MYC cancers, as supported by multiple lines of evidence. In switchable models of MYC-induced lymphoma and osteosarcoma, the ability of MYC to induce ribosomal gene products and enhance protein translation correlates with its ability to sustain tumorigenesis [38]. This has been studied in the *Eμ-MYC* lymphoma model, where MYC dramatically increases global protein synthesis, cell size, cell cycle entry, and lymphomatous transformation [39]. Intercrossing *Eμ-MYC* mice into an L24 ribosomal protein hemizygous background (*RPL24*+/−) blocked MYC’s effects on protein translation and cell size, induced cell cycle entry, and initiated transformation (and increased cell death among MYC-activated lymphoid cells) [39]. Specifically, cap-dependent protein translation was enhanced by deregulated MYC, which persisted through mitosis rather than switching to internal ribosome entry site (IRES)-dependent translation [40]. This coordinated switch normally allows a subset of IRES-dependent mitosis-specific proteins to be translated, and their loss downstream of activated MYC may contribute to genomic instability. This switch was restored in the *RPL24*+/− background or through treatment of *Eμ-MYC* cells with mTOR inhibitors that impair cap-dependent translation. This same *RPL24* haploinsuffient restriction of protein synthesis did not impact oncogenesis in *TP53* null mice, supporting it as an attribute of *MYC* oncogenic signaling [39].

Cap-dependent translation initiation is rate-limiting for most translated proteins and is stimulated by the eukaryotic translation initiation 4F (eIF4F) complex that contains eIF4E that binds the mRNA cap structure, eIF4A, an RNA helicase that prepares the template for ribosome loading, and eIF4G, which provides a scaffold for bridging the mRNA and ribosome pre-initiation complex (reviewed in Lin et al. [41]). MYC stimulates cap-dependent translation through regulation of eIF4E, eIF4A, and eIF4G, while mTORC1 activity also regulates eIF4E. mTORC1-dependent phosphorylation of 4E binding protein-1 (4EBP1), an eIF4E-binding protein, induces the release of eIF4E for translation initiation. Just as backcrossing *Eμ-MYC* mice into a ribosomal protein-insufficient background (*RPL34*+/− or *RPL38*+/−) or treating with an mTORC1 inhibitor like rapamycin reduce MYC-initiated tumorigenesis, so too does backcrossing into a non-phosphorylatable 4EBP1 background in which eIF4E remains inhibited [42]. Thus, MYC exerts both a quantitative and a qualitative control over protein translation to drive oncogenesis.

## 3. Polyamines

Polyamines are small organic cations found in nearly all living organisms that are essential for cell growth and survival. The three principal polyamines synthesized by mammals are putrescine, spermidine, and spermine, and these polycations support cell growth and proliferation through cationic, chaperone-like interactions with anionic macromolecules such as DNA, RNA, proteins, and phospholipids. Polyamines also have sequence-dependent DNA interactions and play roles in specific cellular processes such as DNA methylation, chromatin structure, transcription, ion channel function, and scavenging of reactive oxygen species. While their chaperone functions are important for maximizing protein translation efficiency, polyamines also play a direct role in translation by the absolute requirement for spermidine to activate the eukaryotic translation initiation factor 5A (eIF5A) elongation factor and by indirect regulation of mTORC1 and eIF4E activity. The maintenance of normal intracellular polyamine levels is essential to support cell proliferation [43], and polyamine homeostasis is maintained by a multi-level regulation of their transport, biosynthesis, and catabolism.

### 3.1. Polyamine Biosynthesis and Metabolism

The first, and in most conditions, rate-limiting step in polyamine biosynthesis is the conversion of ornithine (a product of the urea cycle or formed from the catabolism of glutamine) to putrescine catalyzed by ornithine decarboxylase (ODC, encoded by *ODC1*). Putrescine is then converted to spermidine via spermidine synthase (SRM), and spermidine to spermine by spermine synthase (SMS). The activity of the second major regulatory enzyme in polyamine synthesis, adenosylmethionine decarboxylase (AMD, encoded by *AMD1*), provides the aminopropyl groups transferred by SRM and SMS. Both ODC and AMD have the shortest half-lives (10–30 min) of any mammalian enzyme, allowing fine-tuning of polyamine synthesis. Ornithine decarboxylase itself is tightly regulated by transcription, post-transcriptional processing, modified translation efficiency, and altered protein stability, with both antizymes (OAZ1 and OAZ2) and an antizyme inhibitor (AZIN1) contributing [44,45,46,47,48]. Polyamine abundance also negatively feeds back on ODC activity by inducing a +1 ribosomal frameshift that is required to translate functional OAZ1 by reading over a stop codon [49].

Catabolic flux is regulated by the inducible spermidine/spermine-N-acetyltransferase (SAT1) that acetylates higher-order polyamines for export through specific transmembrane solute carriers or for oxidation to lower-order polyamines via polyamine oxidase (PAOX), or by spermine oxidase (SMOX) activities, enhancing homeotic control over the repertoire of cellular polyamines [50]. Finally, an as of yet incompletely characterized energy-dependent polyamine transporter imports polyamines from the microenvironment and can function to rescue polyamine synthesis deficits. These pathways are more completely reviewed in Miller-Fleming et. al. [51]. Indeed, the remarkable investment made to regulate polyamine homeostasis underscores the importance of this pathway in mammalian biology.

### 3.2. MYC, Polyamines, and Cancer

Deregulated polyamine metabolism has been implicated in several pathologies, notably cancer. Polyamines are abundant in proliferative tissues, including cancer tissues, and are present at much lower levels in senescent and non-proliferative tissues [52,53,54]. Neuroblastoma is a highly lethal pediatric malignancy of neural crest cells and a prototypical MYC-driven cancer. *MYCN* is deregulated through genomic amplification in ~40% of high-risk neuroblastomas [9,55], while MYC is frequently deregulated in a high proportion of the remainder [56,57]. Additionally, *ODC1* itself has been shown to be amplified in ~6% of high-risk tumors along with *MYCN* and associates with exceptionally poor tumor survival ([3,58,59] and unpublished data). *ODC1* is a direct MYC target and bona fide oncogene [60,61,62], providing the first demonstration of dual amplification of an oncogenic transcription factor and its oncogenic target gene. Whether this confers an enhanced dependency on polyamines or a resistance mechanism against efforts to deplete polyamines, is under further investigation, since this distinction will inform enrichment strategies for polyamine-depletion cancer trials.

Beyond this somatic activation of polyamine synthesis, transcriptome analyses identified polyamine metabolism as coordinately deregulated by MYC in neuroblastomas. Aggressive tumor behavior and poor outcome is correlated with high expression of synthetic genes (*ODC1*, *SRM*, *SMS*, *AMD1*, *AZIN1*) and low expression of catabolic genes (*OAZ1*, *OAZ2*, *SAT1*, *PAOX*), and these were identified as direct MYC effects by promoter activity and MYCN binding studies ([3]). Many of these correlations retained independent significance in multivariate analyses controlled for key prognostic variables. These data validate polyamine homeostasis as a key oncogenic output of hyperactivated MYC signaling.

### 3.3. Polyamines and Protein Translation

Polyamines are important at both the initiation and elongation stages of mRNA translation, though all of their direct and indirect effects on protein synthesis have yet to be fully elucidated [63]. Studies in fractionated mammalian cell-free translation systems demonstrated that the addition of polyamines stimulated protein translation of diverse mRNA species up to eightfold [64,65]. Spermidine and spermine were more potent in this role than putrescine, and it was noted that some mRNAs appeared particularly dependent on the presence of polyamines for their translation, forming a polyamine modulon [66] or polyamine-dependent translatome.

Polyamines regulate protein translation in cells via effects on translation initiation, by modulation of eIF4F complex activity, and on a specific translation factor, namely, eIF5A. Deficits in these processes likely account for the essential roles of polyamines in cell proliferation [63]. While polyamines function to chaperone other macromolecules, the link between their homeostatic control and protein translation was underscored by studies in which *SAT1* was overexpressed via adenoviral transduction in HeLa cells. Acute hyperactivation of SAT1 cells led to a rapid depletion of cellular spermidine and spermine that coincided with abolished global protein synthesis, without a change in DNA or RNA synthesis [67]. Treatment with difluoromethylornithine (DFMO, an inhibitor of ornithine decarboxylase that depletes putrescine and spermidine from cells) also caused a reduction in global protein synthesis in vitro, further underscoring the connection between polyamines and translation [63]. In the *Eu-MYC* model, treating lymphoma-prone mice with DFMO extended survival [4] to a similar extent as genetic crosses into the *RPL24*+/− background [39] or a dominant-negative 4EBP1 mutant background [42], consistent with the notion that the major mechanism downstream of DFMO is disruption of protein translation. Newer tools now being employed to define the effects of polyamine perturbation on ribosomes and their activities are leading to more refined insights into these pathways.

## 4. Targeting Polyamine Homeostasis and Protein Translation as a Therapy for *MYC*-Driven Cancers

MYC-driven malignancies have a significant dependence on the creation of biomass to support cellular proliferation, specifically on the upregulation of genes that support protein synthesis: ribosome biogenesis, tRNAs, and elongation factors (Figure 1). Indeed, in an analysis across >1000 cancer cell lines from the Cancer Cell Line Encyclopedia, a bioribogenesis and protein translation gene set score was found to be highly correlated with *MYC* expression, with a correlation coefficient of 0.48 at *p* < 0.0001 [68]. This dependence on enhanced protein synthesis for oncogenicity and the concomitant requirement for polyamines to support protein translation provide therapeutic opportunities. Discussed below are strategies to target polyamine metabolism directly as well as key protein translation factors that are influenced by polyamines, as potential points of therapeutic synergy.

### 4.1. Targeting Polyamine Homeostasis

Polyamine metabolism is upregulated in proliferating tissues, including cancers. Increased polyamine production can be inhibited through impairing synthesis, decreasing import, or increasing export. Not surprisingly, compounds affecting each of these facets of polyamine homeostasis, alone or in combination, provide attractive anticancer approaches currently under development.

DFMO, or eflornithine, is an irreversible inhibitor of ornithine decarboxylase. As a modified ornithine analog, DFMO works through enzyme-activated covalent binding to its target enzyme, ODC. Intravenous DFMO is Food and Drug Administration (FDA)- and European Medicines Agency (EMA)-approved for the treatment of Trypanosomiasis (African sleeping sickness), as ODC activity is essential for the survival of that protozoan, and has also shown anti-cancer activity in a variety of preclinical models [3,4,69,70]. DFMO is orally bioavailable and has acceptable tolerability even at high doses, which may be required to inactivate ODC in cells with *MYC* and/or *ODC1* amplification. The mechanisms of DFMO’s anti-cancer activity include tumor cell polyamine depletion and impaired protein translation, although this remains an area of ongoing investigation. DFMO alone significantly delayed tumor initiation in neuroblastoma-prone *TH-MYCN* mice (homozygous for the *MYCN* transgene), although all mice eventually succumbed to the tumor. More surprisingly, initiating DFMO treatment after tumor onset in this rapidly lethal tumor model also extended survival [69]. In that setting, progressing tumors had reduced putrescine, as evidence of ODC inhibition, but preserved spermidine and spermine because of compensatory mechanisms, like activation of AMD and enhanced polyamine uptake to restore polyamine levels, demonstrating that tumor progression downstream of MYC requires sufficient polyamines.

Adenosylmethionine decarboxylase, a second major regulatory enzyme in the polyamine synthetic pathway, has oncogenic activities in prostate cancer where it is stabilized and activated by mTORC1, showing increased levels in mTORC1-activated tumors and decreased levels in tumors from patients treated with the mTORC1 inhibitor everolimus [71]. Pharmacologic inhibition of AMD is achieved with SAM486, and combining this agent with DFMO to inhibit both rate-limiting polyamine synthetic enzymes in *TH-MYCN* homozygous mice was able to prevent tumor initiation in 40% of mice when used pre-emptively, confirming the requirement for polyamine sufficiency downstream of MYC in tumor initiation [69]. However, the same synergistic activity was not seen when DFMO and SAM486 were used to treat established and progressing tumors in this model.

### 4.2. Targeting Polyamine Uptake and Export

Additional efforts to augment polyamine depletion in concert with DFMO therapy include approaches that enhance polyamine export or block uptake mechanisms. Sat1 is the key enzyme regulating the catabolism of the polyamines spermine and spermidine, as this acetyltransferase acetylates them for export from the cell. SAT1 activity can be induced by non-steroidal anti-inflammatory drugs (NSAIDs) such as celecoxib or sulindac, and synergy with DFMO has been shown in preclinical neuroblastoma models where the combination enhanced tumor polyamine depletion and extended survival. More importantly, a randomized, placebo-controlled trial of DFMO in combination with sulindac was shown to markedly reduce the recurrence of adenomas, particularly advanced adenomas, in at-risk adults [2]. Several clinical trials are currently open using DFMO in the treatment of a variety of pre-cancer and cancer states, both alone and in combination with chemotherapeutics and NSAIDs, including in patients with neuroblastoma.

Various polyamine transport inhibitors are being developed for anti-tumor activity in combination with DFMO. AMXT1501 is one such inhibitor that in combination with DFMO has been shown to have anti-tumor and immune stimulatory activities in a mouse model of squamous cell carcinoma [72,73] and also to inhibit neuroblastoma proliferation in vitro [69] and extend survival of *TH-MYCN* mice with neuroblastoma in vivo. Most recently, the polyamine transport inhibitor Trimer44NMe was used in combination with DFMO in an orthotopic mouse model of pancreatic ductal adenocarcinoma, demonstrating tumor growth inhibition as well as decreased MYC expression [74]. Transport inhibitors in combination with polyamine synthesis inhibition may be a promising pharmacologic strategy for the treatment of polyamine-dependent tumors, as this approach blocks the primary rescue pathway utilized by cancer cells undergoing polyamine deprivation stress.

It is possible that inhibiting more than one aspect of polyamine metabolism may provide an effective strategy for targeting the dependency of malignant tissue having high levels of polyamines. Work is ongoing for the transition of several of these polyamine-inhibitory compounds, alone and in various combinations, from a preclinical to clinical setting.

### 4.3. The eIF4F Complex and Cap-Dependent Translation

In addition to synergistically targeting polyamine homeostatic pathways to augment the extent of polyamine depletion, another approach is to look for synergy opportunities in the principal oncogenic pathways disrupted by polyamine depletion. Given the relationship between polyamines and protein translation, this is an area of active investigation.

The eIF4F cap-binding translation initiation complex is responsible for cap-dependent mRNA translation. eIF4F is composed of eF4E (cap-binding protein), eIF4A (ATP-dependent RNA helicase), and eIF4G (scaffolding protein). The formation of the eIF4F complex is highly regulated and dependent on several signaling pathways that are also involved in oncogenesis. eIF4E is the least abundant of the eIF4F complex proteins and therefore is the rate-limiting element in eIF4F formation. 4E-binding proteins sequester eIF4E, and phosphorylation of 4EBP1 by mTORC1 is required for the release of eIF4E. The central role of eIF4E as a highly regulated component of the eIF4F complex and its overexpression in several cancers make eIF4E an attractive target for the development of therapeutic agents [75]. Overexpression of eIF4E transforms immortalized mouse NIH-3T3 cell [76] and cooperates with the viral homolog of MYC to transform rat embryo fibroblasts [77]. Subsequently, eIF4E was found to cooperate with MYC in the *Eμ-MYC* transgenic mouse model to enhance lymphomagenesis and drug resistance in vivo through antagonism of MYC-dependent apoptosis [78,79]. Additionally, induced overexpression of eIF4E leads to the recruitment of ribosomes to a subset of mRNAs that promote oncogenesis, specifically, mRNAs with highly structured 5’ untranslated regions (5’UTRs) [80,81,82]. Of note, mRNAs with highly structured 5’UTRs, that are translated more efficiently in the setting of eIF4E overexpression, include two relevant oncogenes, i.e., *ODC1* and *MYC* [83,84].

Increased MYC expression leads directly to increased transcription of three core components of the eIF4F complex (eIF4E, eIF4AI, and eIF4GI), mediated through canonical E-boxes in their promoters. Enhanced expression of these core components by MYC results in a feed-forward loop with increased eIF4F expression, leading to increased translation of MYC [85]. The polyamine putrescine was found to increase the activation of mTORC1 and the subsequent phosphorylation of 4EBP1, and increasing concentrations of putrescine led to increased protein synthesis and cell proliferation in a dose-dependent fashion. These changes in protein production and proliferation were reversed in the presence of the mTOR inhibitor rapamycin [86]. Further linking mTORC1 with polyamine signaling is its role as an amino acid sensor. Oncogenic polyamine synthesis maintains methionine and s-adenosyl-methionine (SAM) at levels sufficient to inhibit SAMTOR, a nutrient sensor in the mTORC1 pathway, and supports mTORC1 signaling [87]. In the setting of DFMO, there is enhanced AMD activity that leads to reduced pools of methionine and SAM, de-repression of SAMTOR, and inhibition of mTORC1 activation. Additionally, in ODC over-producing cells, there was an increase in eIF4E phosphorylation and subsequent helicase activity of the eIF4F complex [88]. This, in combination with increased efficiency of translation of *ODC1* in the setting of eIF4E overexpression, suggests an additional feed-forward loop of the eIF4F components and an oncogene, here *ODC1*, possibly of relevance to oncogenic transformation.

### 4.4. Targeting The eIF4F Complex

Several agents that inhibit the components of the eIF4F complex have shown preclinical antitumor activity and may have enhanced activity in combination with polyamine depletion agents like DFMO. A 4E-antisense oligonucleotide to reduce the expression of eIF4E was shown to inhibit tumor growth in a prostate cancer xenograft model [89] and was safe in humans in a Phase I clinical trial, though it did not demonstrate anti-tumor activity as a single agent [90]. Pateamine A is a macrolide originally isolated from marine sponges in the 1990s, and an analog of pateamine led to tumor regression in two melanoma mouse xenograft models [91]. Inhibitors of eIF4A helicase activity include hippuristanol, pateamine A, and other flavaglines and rocaglates derived from natural products.

Hippuristanol inhibits the binding of mRNA to eIF4A and has been shown to reverse the resistance to chemotherapy and induce apoptosis in synergy with ABT-737 (BCL-2 inhibitor) in the *Eμ-MYC* lymphoma model [92]. The most extensively studied of the agents targeting the eIF4A helicase is silvestrol, which belongs to the flavagline class and enhances eIF4A binding to mRNA, thus preventing its participation in the eIF4F complex. This agent extended survival when combined with chemotherapy in both the *Eμ-MYC/eIF4E* and *Eμ-MYC*/*PTEN+/−* lymphoma models, in which MYC and mTOR are deregulated to drive lymphoma onset [93]. In the *NOTCH1*-driven murine T-ALL model of acute leukemia, in which *MYC* is also deregulated [94], silvestrol was found to preferentially inhibit the translation of mRNAs with G-quadruplex structures, which include several transcription factors and epigenetic regulators [95].

### 4.5. Eukaryotic Translation Initiation Factor 5A and Its Spermidine-Dependent Activation

Eukaryotic translation initiation factor 5A is a unique RNA-binding protein that is evolutionarily conserved from fungi through plants, insects, and mammals and is essential to eukaryotic cell viability. There are two paralogous genes encoding isoforms of eIF5A: eIF5A1 at chromosome band 17p13 and eIF5A2 at 3q26 [96]. eIF5A1 is ubiquitously expressed, while eIF5A2 expression is largely restricted to the brain and testis. Either isoform can rescue yeast with genetic eIF5A deletion, consistent with their functional redundancy. eIF5A was originally implicated in having stimulatory effects in the formation of the first peptide bond between Met-tRNA and puromycin in translation initiation [97,98]. However, its effects are likely far broader on the basis of polysome analyses demonstrating ribosomal stalling in eIF5A-depleted conditions in yeast [99,100]. The bacterial orthologue, elongation factor P (EF-P), is structurally and functionally similar to eIF5A. EF-P was found to be important in resolving ribosomal stalling at sites of consecutive proline residues [101,102], and eIF5A’s effect on the resolution of polyproline stretches was subsequently established in yeast [103]. Most recently, eIF5A has been shown to have a role in accelerating peptidyl transfer more generally at ribosome stalling sites, though tripeptide motifs have differential dependence on the activity of this factor (with prolines featuring prominently in disruptive motifs), and to promote peptide release from the ribosome at termination [104]. eIF5A appears to effect multiple facets of protein synthesis, and the biologic and therapeutic implications of its functions remain areas of active investigation. The extent to which eIF5A contributes to global protein translation may also be a function of genome complexity, as proline-rich motifs occur at higher frequency in the proteomes of higher organisms [105].

In addition to roles as cationic chaperones, the polyamine spermidine is absolutely required for the activation of eIF5A [106]. eIF5A undergoes a two-step post-translational modification termed hypusination, in which a butyl amine moiety of spermidine is covalently bound to a lysine residue (K50) of eIF5A to yield its hypusinated form (Figure 1). Hypusination of eIF5A is essential for most, if not all, of its activities, therefore making spermidine an essential substrate for eIF5A activation. The two enzymes necessary for hypusination of eIF5A, i.e., deoxyhypusine synthase (DHS) and deoxyhypusine hydroxylase (DOHH), are dedicated exclusively to this pathway and, like eIF5A, are essential genes [107]. No other eukaryotic protein is known to require hypusination, further underscoring the unique importance of this polyamine-dependent modification [108]. Hypusination of eIF5A occurs co-translationally and appears to be irreversible, suggesting a non-dynamic role in regulating eIF5A activity under normal conditions. Inactivation through acetylation by the SAT1 acetyltransferase may provide additional regulation, though this remains incompletely understood [109]. Indeed, while SAT1 can acetylate the hypusine moiety of eIF5A, rendering the factor inactive, prolonged overexpression of SAT1 in HeLa cells abolished protein translation but inactivated less than 10% of the total hypusinated-eIF5A pool [67].

Both eIF5A isoforms have been linked to cancer functions, though, given the weak or undetectable expression of eIF5A2 in most tissues, their overexpression in various cancers is more readily identified. Moreover, high expression of eIF5A2 has been found in numerous cancer in association with *EIF5A2* gene amplification and correlates with advanced cancer stage and worse prognosis for many patients, leading to its identification as an oncogene (reviewed in Nakanishi et al. [110]). However, a targeted shRNA library screen identified *EIF5A*, *AMD1*, *SRM*, and *DHS* as tumor suppressors in the *Eμ-MYC* model [111]. This data has yet to be reconciled with extensive data linking polyamine sufficiency with malignant progression, though it is possible that, in B cells, an eIF5A-dependent translatome includes proteins with tumor suppressor functions disproportionately over those with putative oncogenic roles. Alternatively, there may be functional distinctions between the role of eIF5A in tumor initiation versus maintenance or progression. Indeed, as mentioned, AMD inhibition cooperates with ODC inhibition to block tumor initiation in the neural crest of *TH-MYCN* neuroblastoma-prone mice, yet it antagonizes DFMO activity in the setting of therapy for established and progressing tumors [69].

### 4.6. Inhibition of eIF5A Hypusination

The key enzymes in activating hypusination of eIF5A at lysine K-50 are DHS and DOHH, both of which have direct inhibitors. N1-Guanyl-1,7-diaminoheptane (GC7) inhibits DHS and has been studied in vitro in a variety of cancer cell lines. Indeed, GC7 treatment augments the activity of a variety of chemotherapeutic agents and has synergy with targeted agents such as imatinib in *BCR-ABL*-driven leukemia [112,113,114]. However, the clinical utility of GC7 beyond in vitro studies is uncertain, as it is a spermidine analog and can therefore affect other aspects of polyamine homeostasis not specific to hypusination of eIF5A. DOHH requires iron for the hydroxylation of hypusine, making DOHH susceptible to inhibition by iron chelators such as ciclopirox and deferiprone, both of which are clinically available agents. Ciclopirox is an antifungal agent and has been shown to inhibit growth and stimulate apoptosis in a variety of cancer types in vitro [115,116] and in vivo [117]. Deferiprone is an oral chelator used to treat iron overload secondary to chronic blood transfusion in thalassemia and was found to inhibit proliferation of cervical cancer cells in vitro [118]. A Phase I clinical trial in patients with advance hematologic malignancies to assess dose and side effect profile of ciclopirox demonstrated clinical tolerability of escalated doses and even evidence of disease stabilization in a few patients [119]. It is possible that a combination therapy with one of these agents blocking a key step in the hypusination of eIF5A and a polyamine antagonist may show additive or synergistic anti-tumor effects.

## 5. Conclusions: Current Gaps and Future Perspective

MYC-driven bioribogenesis and protein synthesis are central to its oncogenicity. While directly targeting MYC has proven difficult, downstream effectors driving increases in global and selective protein synthesis provide numerous targetable pathways that have potent oncogenic roles and an exploitable therapeutic index. Here, we have highlighted some of these pathways that are essential to the increased protein synthetic drive induced by MYC. Though polyamine homeostasis is essential for normal cell growth and proliferation, an increased drive for polyamine uptake and synthesis noted in human cancers may provide a targetable MYC-driven dependency. Gene expression signatures of MYC-driven cancers often demonstrate increased expression of enzymes involved in polyamine synthesis and decreased expression of polyamine catabolic enzymes, highlighting this coordinated signaling downstream of MYC. Polyamines play an important role in protein synthesis, both globally and selectively, by activating key translation factors such as eIF5A. However, the exact mechanisms by which polyamine depletion leads to protein translation stress and the mechanisms of anti-tumor activity when polyamines are depleted from tumor cells remain the subject of ongoing work. 

Current studies in our laboratory and others are aimed at using a combination strategy to induce protein translation stress through direct polyamine depletion in addition to inhibition of the translation machinery in neuroblastoma and other MYC-driven tumor models. Polyamine depletion, alone or in combination with inhibitors of protein translation, may provide a potent stress on a MYC-driven cancer, which may result in clinically significant anti-tumor activity. Such preclinical work is required for the informed design of clinical protocols to test these approaches, with the hope of understanding which additional compounds, if any, provide the highest likelihood of clinically relevant anti-tumor effects. 

## Figures and Tables

**Figure 1 medsci-06-00041-f001:**
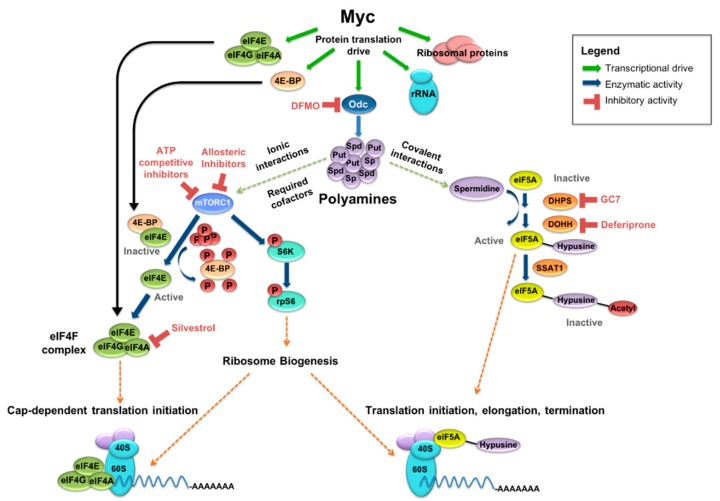
MYC drives protein synthetic output and polyamine metabolism. MYC-driven output includes ribosomal proteins, rRNA, and translation factors, among others. MYC also transcriptionally upregulates ornithine decarboxylase (ODC), a rate-limiting enzyme in polyamine synthesis, as well as spermidine synthase (SRM) and other key polyamine enzymes. Polyamines support protein synthesis in several ways, including spermidine-dependent activation of eukaryotic initiation factor 5A (eIF5A) and as cofactors in the mTORC1-driven release of eIF4E from eIF4E-binding protein 1 (4EBP1) and phosphorylation of S6K. Inhibitors exist for several key enzymes in these polyamine-supported pathways, allowing for the possibility of indirect MYC inhibition through a multi-faceted pharmacologic approach.

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
