# Peer review of "Myc, Oncogenic Protein Translation, and the Role of Polyamines"

_medsci, 2018, doi:10.3390/medsci6020041_

Reviewer 1 Report

The manuscript describes Myc-driven regulation of polyamines and protein synthetic capacity as a key function of its oncogenic capacity and an ongoing approach in targeting these downstream pathways, namely, translation machinery and the polyamine pathway in the control of Myc-driven cancers. Considering difficulties/complexities in targeting Myc directly, these downstream pathways of Myc hold promise as targets in the control of Myc-driven cancers. The review provides a comprehensive coverage of relevant, updated literatures on the topics and an in-depth discussion and is clearly written. It should be acceptable for publication in Medical Sciences if the minor points (listed below) are addressed.

 Minor points

1.       Line 41, “they have been difficult to directly inhibit…”. : Since there have been a number of studies, including targeting Myc-Max dimerization, Myc-Max DNA binding, small drug-like molecules, microRNAs, antisense RNA, it would be informative to mention these attempts (without details).

2.       Lines 79 and 144: Would  “leads to”  be better than “encodes”?

3.       Line 108 “chaperone-like interaction”: Polyamines, in general, bind to DNA and RNA due to their polycationic charges (simple ionic interaction) except for a few specific interactions (sequence-dependent specific sequences). In this regard, “ chaperon-like interaction” may be misleading.

4.       Line 124: OAZ3 is a testis specific antizyme and does not bind to or facilitate degradation of ODC. OAZ3 should be deleted.

5.       Line 127: “by spermine oxidase”  “by” should be inserted, because SMOX is not a part of SAT1-PAOX pathway.

6.       Line 150: “SMOX” is not a polyamine biosynthetic gene and should be deleted.

7.       Line 171: “that depletes polyamines” Here, two different studies, i) the depletion of spermidine and spermine by SAT1 overexpression and ii) depletion of putrescine and spermidine (different depletions in two cases) by DFMO are compared. So changing “polyamines” to “ putrescine and spermidine” would be better for clarification.

8.       Line 171: “causes a similar reduction”.  The reduction in protein synthesis in SAT1-overexpressing cells is much more extensive than in DFMO-treated cells. So changing to “ also causes a reduction” would be better.

9.       Line 270: “recruitment of polyribosomes to s subset”  Ribosomes are recruited to a mRNA or a mRNA-polyribosome complex. Change “polyribosome” to “ribosome”.

10.   Line 278: “ increased transcription of MYC”. Increased eIF4E leads to “increased translation of Myc”?

11.   Line 325-326: “peptidyl transfer at most amino acid sequences” eIF5A relieves ribosome stalling not only at polyproline stretches, but more generally at other ribosome stalling sites. As not most of the amino acid sequences cause ribosome stalling, change “most amino acid sequence” to “generally at ribosome stalling sites”.

12.   Line 334: “spermidine moiety is…”. “a butyl amine moiety of spermidine” is correct.

13.   Line 336: “therefore” a typo

14.   Line344: “prolonged hyperactivation” “prolonged overexpression” is correct.

15.   Line 398: “to inform the design”  Is “for the informed design” better?

Author Response

Thank you for the comments. See changes made as per below. 

1.       Line 41, “they have been difficult to directly inhibit…”. : Since there have been a number of studies, including targeting Myc-Max dimerization, Myc-Max DNA binding, small drug-like molecules, microRNAs, antisense RNA, it would be informative to mention these attempts (without details). Added a sentence including these attempts

2.       Lines 79 and 144: Would  “leads to”  be better than “encodes”? Changed

3.       Line 108 “chaperone-like interaction”: Polyamines, in general, bind to DNA and RNA due to their polycationic charges (simple ionic interaction) except for a few specific interactions (sequence-dependent specific sequences). In this regard, “ chaperone-like interaction” may be misleading. Changed wording

4.       Line 124: OAZ3 is a testis specific antizyme and does not bind to or facilitate degradation of ODC. OAZ3 should be deleted.Done

5.       Line 127: “by spermine oxidase”  “by” should be inserted, because SMOX is not a part of SAT1-PAOX pathway. Done

6.       Line 150: “SMOX” is not a polyamine biosynthetic gene and should be deleted. Deleted

7.       Line 171: “that depletes polyamines” Here, two different studies, i) the depletion of spermidine and spermine by SAT1 overexpression and ii) depletion of putrescine and spermidine (different depletions in two cases) by DFMO are compared. So changing “polyamines” to “ putrescine and spermidine” would be better for clarification.Changed as recommended

8.       Line 171: “causes a similar reduction”.  The reduction in protein synthesis in SAT1-overexpressing cells is much more extensive than in DFMO-treated cells. So changing to “ also causes a reduction” would be better. Changed

9.       Line 270: “recruitment of polyribosomes to s subset”  Ribosomes are recruited to a mRNA or a mRNA-polyribosome complex. Change “polyribosome” to “ribosome”. Changed

10.   Line 278: “ increased transcription of MYC”. Increased eIF4E leads to “increased translation of Myc”? Changed

11.   Line 325-326: “peptidyl transfer at most amino acid sequences” eIF5A relieves ribosome stalling not only at polyproline stretches, but more generally at other ribosome stalling sites. As not most of the amino acid sequences cause ribosome stalling, change “most amino acid sequence” to “generally at ribosome stalling sites”. Changed

12.   Line 334: “spermidine moiety is…”. “a butyl amine moiety of spermidine” is correct. Changed

13.   Line 336: “therefore” a typo Fixed

14.   Line344: “prolonged hyperactivation” “prolonged overexpression” is correct. Changed

15.   Line 398: “to inform the design”  Is “for the informed design” better? Changed

Reviewer 2 Report

Comments for Authors: medsci-296373

 Flynn and Hogarty provide an interesting and comprehensive review of the control of protein translation by Myc family oncoproteins and how this is influenced by their coordinated transcriptional control of enzymes that direct polyamine biosynthesis. This is a particularly interesting topic given the established indirect and direct roles of polyamines in controlling translation, which as the authors review is a therapeutic opportunity, along with strategies that target components of the translational machinery such as eIF5A and the eIF4F complex.

 Overall the review is comprehensive and insightful, and it should be highly read by those in the field. In a few places the text was rather awkward and suggestions for revisions are provided below. In addition, there are a few missing facts the authors should consider including in their review, and they need to add a section regarding polyamine-directed ribosomal frameshifting.

 1. Lines 35-37, suggest revising this sentence to read “MYC plays a central role in creating the biomass necessary to drive cell cycle progression, including significant increases in protein translation”.

 2. Lines 40-41. MYC oncoproteins (which should not be italicized) also provide an attractive target for cancer therapy due to the fact that tumors driven by MYC are addicted to these oncoproteins – see work of Felsher and colleagues. This fact should be noted.

 3. Line 45. Suggest deleting “oncogenic protein”

 4. Line 50. Change “alteration” to “alterations”

 5. Line 51, the claim that MYC oncoproteins are activated in the majority of human cancers is too strong. A more realistic number is half of all human tumor types.

 6. Lines 57 and 266 – the term “murine” refers to the subfamily Murinae, which also includes rats; therefore, suggest changing this to “mouse cancer models” and “mouse NIH 3T3 cells."

 7. Line 77. Suggest revising this to read: “…importance of proteogenesis to MYC-driven oncogenesis”.

 8. Line 90. Suggest replacing “mitotic” with “genomic”

 9.  Lines 84, 90 and 101. “RPL24 +/-“ should be denoted as RPL24+/-. Same is true for RPL38+/- on line 101.

 10. Line 108. The functions of polyamines in regulating cell growth (mass) and proliferation are much broader than the “chaperone-like interactions: noted here where there are also published roles of polyamines in controlling chromatin structure, replication, DNA methylation, transcription, mRNA half-life, and ion channel function, as well as having roles as scavengers of reactive oxygen species (ROS).

 11. Line 116. Ornithine is not simply a product of the urea cycle as stated here, as it can also be produced via the catabolism of glutamine. Indeed, this is the major source of ornithine in some cells.

 12.   Section 3. A paragraph describing the regulatory role of polyamines in controlling the regulatory frameshifting of Antizyme (Oaz1) is called for, where Antizyme mRNA translation depends upon a polyamine-stimulated +1 ribosomal frameshift.

 13. Line 145 – suggest replacing “encodes” with “confers”

 14. Line 165 – I am not sure “essentiality” is a word. Suggest revising this to read “…likely account for the essential roles of polyamines in cell proliferation”.

 15. Line 170. Revise to read “Treatment with difluoromethylornithine…”

 16. Legend to Figure 1. Myc also transcriptionally upregulates spermidine synthase (Forshell et al., 2010, PMID 20103729).

 17. Line 199 – not clear to the reader what Increase polyamine drive is. Perhaps replace drive with “production”?

 18. Line 247, replace “the approach” with “this approach”

 19. Line 250, replace “on” with “having”

 20. Line 267. The term v-Myc is introduced here without any explanation.

 21. Line 278. Is there a mistake here? Shouldn’t “leading to increased transcription” be “leading to increased translation”?

 22.  Section 4.3. It is unclear how putrescine augments activation of mTORC1. Are there data that address mechanism?

 23. Line 284, suggest changing “maintained” to “maintaining” and “to levels that are sufficient”. Also the term/regulator Samtor is never defined or explained.

 24. Line 314. Delet “highly”

 25. Line 336. “therefor” should be “therefore”

 26. Lines 364-367, suggest revising this  sentence to read “Indeed, GC7 treatment augments the activity of a variety of chemotherapeutic agents, and has synergy with targeted agents such as imatinib in BCR-ABL-driven leukemia [106-108].”

27. Section 4.6. Here it should be noted that DOHH coordinates iron and that hydroxylation of hypusine is therefore iron dependent and susceptible to inhibition by iron chelators such as Deferiprone.

 Author Response

1. Lines 35-37, suggest revising this sentence to read “MYC plays a central role in creating the biomass necessary to drive cell cycle progression, including significant increases in protein translation”. Changed

 2. Lines 40-41. MYC oncoproteins (which should not be italicized) also provide an attractive target for cancer therapy due to the fact that tumors driven by MYC are addicted to these oncoproteins – see work of Felsher and colleagues. This fact should be noted.Added

 3. Line 45. Suggest deleting “oncogenic protein” Changed

 4. Line 50. Change “alteration” to “alterations” Changed

 5. Line 51, the claim that MYC oncoproteins are activated in the majority of human cancers is too strong. A more realistic number is half of all human tumor types. Changed

 6. Lines 57 and 266 – the term “murine” refers to the subfamily Murinae, which also includes rats; therefore, suggest changing this to “mouse cancer models” and “mouse NIH 3T3 cells." Changed

 7. Line 77. Suggest revising this to read: “…importance of proteogenesis to MYC-driven oncogenesis”. Changed

 8. Line 90. Suggest replacing “mitotic” with “genomic” Changed

 9.  Lines 84, 90 and 101. “RPL24 +/-“ should be denoted as RPL24+/-. Same is true for RPL38+/- on line 101. Changed

 10. Line 108. The functions of polyamines in regulating cell growth (mass) and proliferation are much broader than the “chaperone-like interactions: noted here where there are also published roles of polyamines in controlling chromatin structure, replication, DNA methylation, transcription, mRNA half-life, and ion channel function, as well as having roles as scavengers of reactive oxygen species (ROS). Re-worded

 11. Line 116. Ornithine is not simply a product of the urea cycle as stated here, as it can also be produced via the catabolism of glutamine. Indeed, this is the major source of ornithine in some cells. Added

 12.   Section 3. A paragraph describing the regulatory role of polyamines in controlling the regulatory frameshifting of Antizyme (Oaz1) is called for, where Antizyme mRNA translation depends upon a polyamine-stimulated +1 ribosomal frameshift. Added

 13. Line 145 – suggest replacing “encodes” with “confers” Changed

 14. Line 165 – I am not sure “essentiality” is a word. Suggest revising this to read “…likely account for the essential roles of polyamines in cell proliferation”. Changed

 15. Line 170. Revise to read “Treatment with difluoromethylornithine…” Changed

 16. Legend to Figure 1. Myc also transcriptionally upregulates spermidine synthase (Forshell et al., 2010, PMID 20103729).  Didn’t alter figure for simplicity sake

 17. Line 199 – not clear to the reader what Increase polyamine drive is. Perhaps replace drive with “production”? Changed

 18. Line 247, replace “the approach” with “this approach” Changed

 19. Line 250, replace “on” with “having” Changed

 20. Line 267. The term v-Myc is introduced here without any explanation. Added comment

 21. Line 278. Is there a mistake here? Shouldn’t “leading to increased transcription” be “leading to increased translation”? Changed

 22.  Section 4.3. It is unclear how putrescine augments activation of mTORC1. Are there data that address mechanism? Added

 23. Line 284, suggest changing “maintained” to “maintaining” and “to levels that are sufficient”. Also the term/regulator Samtor is never defined or explained. Changed

 24. Line 314. Delet “highly” Changed

 25. Line 336. “therefor” should be “therefore” Changed

 26. Lines 364-367, suggest revising this  sentence to read “Indeed, GC7 treatment augments the activity of a variety of chemotherapeutic agents, and has synergy with targeted agents such as imatinib in BCR-ABL-driven leukemia [106-108].” Changed

 27. Section 4.6. Here it should be noted that DOHH coordinates iron and that hydroxylation of hypusine is therefore iron dependent and susceptible to inhibition by iron chelators such as Deferiprone. Changed

Reviewer 3 Report

This manuscript by Flynn and Hogarty provides an excellent review of the role of polyamines in mammalian translation with special emphasis on this mechanism in human cancers. The paper will be of interest to readers interested in protein synthesis in eukaryotes and to those with more clinical interests in human cancers.

 Prior to publication, the authors should consider several items for revision/clarification.

 1.     L94: The statement that cap-dependent translation is rate-limiting for translation is a bit strong and should recognize the importance of cap-independent translation in certain situations (e.g. myotonic dystrophy type 2; Plos One vol 5 issue 2, 2010)

2.     L106: Suggest a more nuanced definition of the “three” polyamines. Cadavarine (another polyamine) can be synthesized in mammalian cells expressing high levels of ODC via abnormal lysine decarboxylation.

3.     L116: Suggest caution and consistency in use of the term “rate-limiting.” First enzyme or second step, for example, may be more precise terms. The use of “rate-limiting” and then “second rate-limiting” is confusing. It is true that one or the other can be rate-limiting, but not both in the same conditions.

4.     L155: sp. “Poylamines”

5.     L204: A high dose IV form of DFMO was approved by both the FDA and EMA for the African Sleeping Sickness indication, but no oral formulation of DFMO has ever gained regulatory approval for any indication.

6.     L233-235: Reference 2 does not show that DFMO in combination with sulindac prevents “… malignant transformation from colonic adenomas to adenocarcinoma in at-risk adults..” Reference 2 does describe a trial to evaluation the efficacy of this combination to prevent the development of metachronous colorectal adenomas in patients with sporadic risk of these precancers. A subsequent paper Meyskens et al (2008) Cancer Prevention Research volume 1 describes a trial in which this combination was shown to be remarkably effective in this regard.

Author Response

Thank you for your review. Please note changes acknowledged in red per below. 

1.     L94: The statement that cap-dependent translation is rate-limiting for translation is a bit strong and should recognize the importance of cap-independent translation in certain situations (e.g. myotonic dystrophy type 2; Plos One vol 5 issue 2, 2010). Changed the wording of this so as to not imply all protein translation is cap-dependent. 

2.     L106: Suggest a more nuanced definition of the “three” polyamines. Cadavarine (another polyamine) can be synthesized in mammalian cells expressing high levels of ODC via abnormal lysine decarboxylation. Changed to make for a broader interpretation of the canonical polaymines

3.     L116: Suggest caution and consistency in use of the term “rate-limiting.” First enzyme or second step, for example, may be more precise terms. The use of “rate-limiting” and then “second rate-limiting” is confusing. It is true that one or the other can be rate-limiting, but not both in the same conditions. Removed rate-limiting

4.     L155: sp. “Poylamines” Fixed

5.     L204: A high dose IV form of DFMO was approved by both the FDA and EMA for the African Sleeping Sickness indication, but no oral formulation of DFMO has ever gained regulatory approval for any indication. Clarified that intravenous form is FDA/EMA approved

6.     L233-235: Reference 2 does not show that DFMO in combination with sulindac prevents “… malignant transformation from colonic adenomas to adenocarcinoma in at-risk adults..” Reference 2 does describe a trial to evaluation the efficacy of this combination to prevent the development of metachronous colorectal adenomas in patients with sporadic risk of these precancers. A subsequent paper Meyskens et al (2008) Cancer Prevention Research volume 1 describes a trial in which this combination was shown to be remarkably effective in this regard. Changed to the Meyskens reference

Reviewer 4 Report

This review article is both well written and adequately structured, offering a synthetic and to the point view of the relevance of polyamine metabolism and its interplay with MYC proteins related to the control of protein synthesis, in the context of cancer development, and the opportunities for pharmacological inhibition of this pathways with clinical relevance. 

Some minor comments to the authors: 

-Please, check the text to detect and correct misspellings like in line 155 (Poylamines), line 311 (regulars) or line 336 (therefor). 

- Please, check the consistency of the use of capitals or lower case for the names of proteins. It is recommended to always use capitals when talking about the human proteins (MYC, MYCN, ODC, etc) and just first capital and the rest in lower case for murine proteins (Myc, Mycn, Odc). Capitals have only been used for DHS and DOHH. 

- It would be interesting and clarifying to include in Fig 1 the positive feedback loops described in the text, or add a second figure to describe these loops. 

- A reference to Fig 1 should be included in the main text when describing hypusination. 

- At the beginning of section 2.1, please clarify that MYC has been described both as a general transcriptional amplifier with low affinity and as an transcription factor controlling the expression of specific genes leading to increased RNA synthesis, and cite the corresponding papers (works from Young's and Levens's labs in Cell, 2012, and from Amati's and Eiler's labs, Nature, 2014). 

Author Response

-Please, check the text to detect and correct misspellings like in line 155 (Poylamines), line 311 (regulars) or line 336 (therefor).  Changed

- Please, check the consistency of the use of capitals or lower case for the names of proteins. It is recommended to always use capitals when talking about the human proteins (MYC, MYCN, ODC, etc) and just first capital and the rest in lower case for murine proteins (Myc, Mycn, Odc). Capitals have only been used for DHS and DOHH. Changed

- It would be interesting and clarifying to include in Fig 1 the positive feedback loops described in the text, or add a second figure to describe these loops. Did not alter figure for simplicity sake

- A reference to Fig 1 should be included in the main text when describing hypusination. Added

- At the beginning of section 2.1, please clarify that MYC has been described both as a general transcriptional amplifier with low affinity and as an transcription factor controlling the expression of specific genes leading to increased RNA synthesis, and cite the corresponding papers (works from Young's and Levens's labs in Cell, 2012, and from Amati's and Eiler's labs, Nature, 2014).  Sentences added in this regard